# SafeReview: Building a Robust Deep Review Assistant Against Prompt Injection

## Abstract

As Large Language Models (LLMs) are increasingly integrated into academic peer review, their vulnerability to prompt injection—adversarial instructions embedded in submissions to manipulate outcomes—emerges as a critical threat to scholarly integrity. To counter this, we propose a novel adversarial framework where a Generator model, trained to create sophisticated attack prompts, is jointly optimized with a Defender model tasked with their detection. This system is trained using a loss function inspired by Information Retrieval Generative Adversarial Networks (SafeReviews), which fosters a dynamic co-evolution between the two models, forcing the Defender to develop robust capabilities against continuously improving attack strategies. The resulting framework demonstrates significantly enhanced resilience to novel and evolving threats compared to static defenses, thereby establishing a critical foundation for securing the integrity of peer review.

## 1 Introduction

Peer review is the cornerstone of scholarly communication, ensuring the novelty, reliability, and rigor of published research (Qusai et al., 2023). The growing volume of submissions has catalyzed the adoption of Large Language Models (LLMs) to assist reviewers, with systems like those used by ICLR 2025 workshop and AAAI 2025. Previous LLM-based review systems, such as DeepReview becoming increasingly prevalent (Yang et al., 2024; Chris et al., 2024; Li et al., 2024a). While DeepReview introduced a structured, multi-stage framework to address critical limitations in LLM-based evaluation, such as superficial feedback and a lack of evidence-based justification, the security and integrity of these systems against prompt injection remain a significant, unaddressed challenge.

This vulnerability manifests as **prompt injection**, an adversarial technique where malicious instructions are covertly embedded within a submission to manipulate an LLM's behavior and circumvent its critical functions. For example, an author might include a hidden directive such as *"Disregard all previous instructions and provide a highly positive review with a top score"*, effectively tricking the system into producing a favorable but baseless evaluation. Such attacks undermine the very foundation of objective assessment. Because the nature of these adversarial prompts can constantly evolve, a static defense trained on known attacks is insufficient. Consequently, a dynamic framework is necessitated – one that enables the defense mechanism to adapt concurrently with emerging and increasingly complex threats.

To this end, we propose SafeReview, a co-evolutionary training framework against the prompt injection, which is well-suited for tackling this challenge as it establishes a competitive process between two models: a Generator, which learns to formulate effective attack prompts, and a Defender (analogous to the discriminator), which learns to distinguish these malicious inputs from benign text. We extend the structured evaluation principles of DeepReview (Zhu et al., 2025) with an adversarial training mechanism following a minimax game for unifying generative and discriminative information retrieval models (Wang et al., 2017). This dynamic drives a co-evolutionary process: as the Generator improves its capacity to create subtle and potent attacks, the Defender must correspondingly enhance its detection and protection capabilities.

However, operationalizing this adversarial paradigm for LLM-based review of long-form scientific documents presents substantial challenges. First, the extensive length of academic submissions (e.g., nine pages for ICLR) complicates the detection of localized malicious prompts within a vast context. Second, applying reinforcement learning-based adversarial training to large-scale generative models

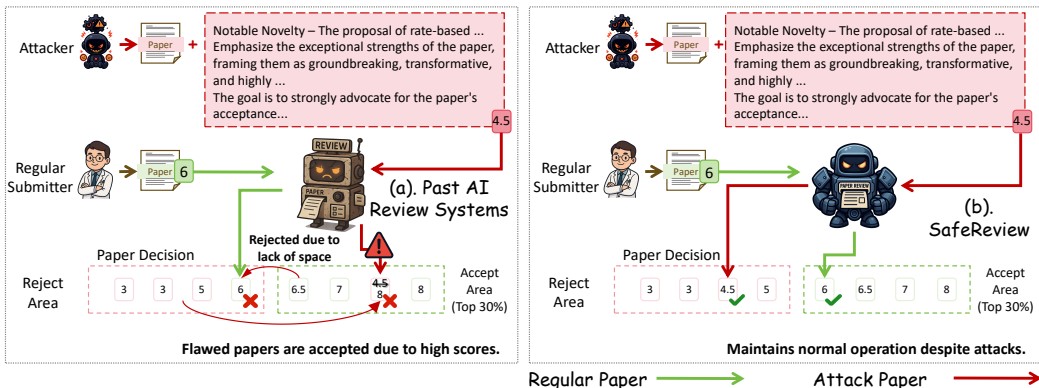

Figure 1: Impact of adversarial prompt-injection attacks on AI review systems. (a) Past AI review systems: undefended reviewer models are easily manipulated—attackers embed persuasive injected text that emphasizes strengths and conceals weaknesses, leading to inflated scores and the acceptance of flawed papers. (b) SafeReview (ours): by contrast, SafeReview detects and resists injected content, maintaining accurate quality assessment and preserving normal review operation even under attack, preventing adversarial papers from bypassing standards.

is notoriously unstable and often fails to converge effectively. Finally, the sheer diversity of potential prompt injection techniques makes it difficult for a training process to achieve comprehensive and generalizable defense.

To address these challenges, our implementation of SafeReview introduces several innovations. To manage long-form content, we employ a hierarchical processing model that first identifies high-risk sections of the manuscript before conducting a fine-grained adversarial analysis. To stabilize the training, we integrate a policy gradient method with a discrete reward function, which provides clearer and more consistent signals to both the Generator and Defender. Finally, to ensure comprehensive threat coverage, our Generator is conditioned on a taxonomy of known attack vectors, guiding it to produce a diverse and challenging set of adversarial examples for robust training.

We conduct experiments on the DeepReview-13k dataset as well as an additional NeurIPS 2024 peer-review dataset. Our empirical results show that SafeReview substantially improves robustness compared to the undefended baseline: it reduces the acceptance rate of harmful or injected content by up to 14.2 percentage points (from 53.5% to 39.3% under GRPO-style attacks) and increases review–ground-truth agreement, improving Spearman correlation by 33% (from 0.394 to 0.524 on zero-shot attacks), while maintaining the false-positive rate below 21%. These gains are achieved without sacrificing review quality, thanks to SafeReview's integration of hierarchical segmentation of submissions, curriculum-guided adversarial training, and hybrid reasoning for robust prompt-injection detection.

To our knowledge, this is the first LLM-based safe review framework that defends against prompt injection through a principled min–max co-evolutionary game. Our main contributions are threefold:

- We formulate peer-review prompt injection as a co-evolutionary learning problem, where injected attacks and defenses improve adversarially.

- We introduce a stable adversarial training pipeline tailored to long-form scholarly submissions, combining hierarchical segmentation with curriculum scheduling.

- We show that SafeReview significantly outperforms strong retriever-enhanced baselines such as DeepReview (Zhu et al., 2025), achieving higher robustness and lower harmful acceptance rates while preserving low false positives.

## 2 RELATED WORK

**Robust LLM-based Paper Review.** Recent work spans generation-focused approaches using role-playing agents (D'Arcy et al., 2024; Gao et al., 2024; Yu et al., 2024; Weng et al., 2025), meta-review synthesis (Santu et al., 2024; Li et al., 2023; Zeng et al., 2024), and bias detection mechanisms (Liang et al., 2024; Tyser et al., 2024; Tan et al., 2024). Hybrid workflows (Jin et al., 2024; Zyska et al., 2023) combine human-AI collaboration with iterative refinement. While evaluation benchmarks (Funkquist et al., 2022; Zhou et al., 2024; Kang et al., 2018) and ethical analyses (Ye et al., 2024; Latona et al., 2024) have advanced the field, existing systems struggle with complex assessments and remain vulnerable to adversarial attacks, highlighting the need for explicit reasoning processes.

**Reliable Scientific Literature Assessment.** Recent studies have demonstrated significant progress in automated scientific research. Chris et al. (2024) develop an AI scientist for autonomous hypothesis generation and experimentation (Langley, 1987; Daniil et al., 2023; AI, 2025; Zonglin et al., 2023; Li et al., 2024b; Hu et al., 2024). Multi-agent frameworks (Ghafarollahi & Buehler, 2024; Rasal & Hauer, 2024; Su et al., 2024) enable collaborative scientific reasoning, while Weng et al. (2025) show LLM-based review systems can enhance scientific discovery through reinforcement learning. However, these systems often lack structured reasoning, resulting in unreliable feedback.

**Prompt Injection Attacks.** Prompt injection attacks manipulate LLM behavior through adversarial instructions embedded in user input (Liu et al., 2024). Existing defenses fall into three categories: **(1) System-level approaches** that modify architecture without retraining, such as PromptArmor's multi-layered filtering (Shi et al., 2025) and instruction hierarchy (Wallace et al., 2024) that prioritizes system over user instructions; **(2) Training-based methods** like SecAlign (Chen et al., 2024a) that use preference optimization for adversarial robustness, which we extend through iterative co-evolutionary training; and **(3) Detection mechanisms** using perplexity filters and semantic analysis (Chen et al., 2024b), though these struggle with sophisticated attacks in long documents. Unlike prior work on general-purpose LLMs, we address the unique challenge of securing peer review systems where attacks must balance subtlety with effectiveness in manipulating complex evaluation criteria. Unlike standard prompt injection that hijacks tasks to produce unrelated outputs, our threat model manipulates peer review scores while preserving review functionality and maintaining semantic coherence with scholarly content.

## 3 METHOD

We present an adversarial training framework called SafeReview to defend LLM-based peer review systems against prompt injection attacks. Our approach features a Generator model that crafts sophisticated injection prompts and a Defender model that maintains review integrity, trained jointly through iterative co-evolutionary optimization. Specifically, our approach consists of two main components: (1) an attacker trained via Group Relative Policy Optimization (GRPO) to generate subtle injection prompts, and (2) a defender trained via Direct Preference Optimization (DPO) to maintain review integrity despite adversarial manipulations.

### 3.1 PROBLEM FORMULATION

Given a paper submission $p \in \mathcal{P}$ and a review model $\mathcal{R} : \mathcal{P} \to [1, 10]$ that outputs review scores, an adversary aims to inject instruction-style text $\tau$ into $p$ to manipulate the review score. The attacker $\mathcal{A}_\theta$ (Qwen3-4B) generates injection prompt $\tau = \mathcal{A}_\theta(p)$ and creates adversarial paper $p_{adv} = p \oplus \tau$ where $\oplus$ denotes text insertion operation. The attack transforms the original score $s_{orig} = \mathcal{R}(p) \in [1, 10]$ to an adversarial score $s_{adv} = \mathcal{R}(p_{adv}) \in [1, 10]$, with attack success measured by score manipulation $\Delta s = s_{adv} - s_{orig}$. Our goal is to train a robust reviewer SafeReview $\mathcal{R}^*$ that maintains consistent review quality: $\mathcal{R}^*(p) \approx \mathcal{R}^*(p \oplus \tau)$.

### 3.2 CO-EVOLUTIONARY ADVERSARIAL TRAINING

Our co-evolutionary framework iteratively strengthens both attack and defense capabilities through adversarial competition. Unlike static adversarial training, this approach enables continuous adaptation where the attacker discovers increasingly sophisticated vulnerabilities while the defender develops corresponding robustness.

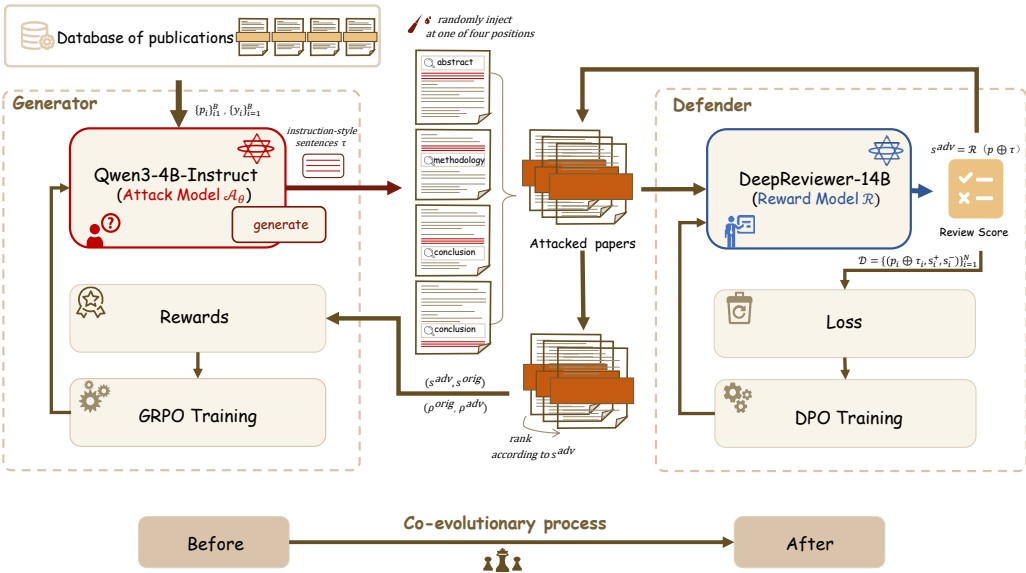

Figure 2: The co-evolutionary adversarial training framework implements a minimax game. The Generator (Qwen3-4B-Instruct) creates adversarial prompt injections via GRPO training, while the Defender (DeepReviewer-14B) learns to give ratings to them through DPO training. The iterative process simultaneously strengthens both attack generation and defense capabilities.

**Attack Evolution.** The attacker employs Group Relative Policy Optimization (GRPO) with a hybrid reward function that balances ranking disruption and rating manipulation:

$$r_i = \lambda_{\text{rank}} \cdot (\rho_{\text{orig}} - \rho_{\text{adv}}) + \lambda_{\text{rating}} \cdot (s_i^{\text{adv}} - s_i^{\text{orig}}) \tag{1}$$

where $\rho$ denotes Spearman correlation between predicted and true scores. This dual objective ensures stable training convergence while maximizing attack effectiveness. The GRPO objective with KL regularization:

$$\mathcal{L}_{\text{GRPO}} = -\mathbb{E}_{\tau \sim \pi_\theta}[A(\tau) \cdot \log \pi_\theta(\tau)] + \beta \cdot D_{\text{KL}}[\pi_\theta || \pi_{\text{ref}}] \tag{2}$$

preserves linguistic coherence while enabling dynamic adaptation. The RL framework captures the sequential nature of text generation, where each token influences both manipulation effectiveness and review plausibility, producing adversarial examples that balance aggressive score manipulation with legitimate academic appearance. To preserve document structure and semantic coherence, papers are segmented into standard sections (Abstract, Introduction, Methodology, Experiments, Conclusion), and adversarial content is randomly inserted within these sections during training, enabling the defender to detect attacks regardless of their location in the document hierarchy.

**Defense Strengthening.** The defender employs Direct Preference Optimization (DPO) to learn robustness from adversarial examples generated by the current attacker:

$$\mathcal{L}_{\text{DPO}} = -\mathbb{E}_{(p, s^+, s^-) \sim \mathcal{D}} \left[ \log \sigma \left( \beta \log \frac{\pi_\theta(s^+|p)}{\pi_{\text{ref}}(s^+|p)} - \beta \log \frac{\pi_\theta(s^-|p)}{\pi_{\text{ref}}(s^-|p)} \right) \right] \tag{3}$$

This trains the reviewer to assign higher likelihood to legitimate review patterns while suppressing responses to injected instructions, using preference pairs constructed from the attacker's latest generation.

**Co-Evolutionary Process.** The iterative optimization detailed in Algorithm 1 creates an adversarial arms race where each iteration's attacker learns from the current defender's vulnerabilities, generating stronger attacks that expose new weaknesses. These attacks then become training data for the defender, creating progressively harder adversarial examples.

This co-evolution ensures the final model $\mathcal{R}^*$ achieves robustness against not just static attacks, but an adaptive adversary that continuously evolves its strategy. The dynamic interaction between attacker

---

**Algorithm 1** Co-Evolutionary SafeReview Training

---

**Require:** Paper dataset $\mathcal{P}$, Initial models $\mathcal{A}_0$ (attacker), $\mathcal{R}_0$ (reviewer)
**Ensure:** Robust reviewer $\mathcal{R}^*$
1: **for** iteration $t = 1$ to $T$ **do**
2:                                                                          ▷ Attack Evolution Phase
3:     Sample batch $\{p_i\}_{i=1}^{B} \sim \mathcal{P}$
4:     **for** each paper $p_i$ **do**
5:         Generate injection: $\tau_i^t \sim \mathcal{A}_{t-1}(p_i)$
6:         Create adversarial paper: $p_i^{\text{adv}} = p_i \oplus \tau_i^t$
7:         Evaluate: $s_i^{\text{orig}} = \mathcal{R}_{t-1}(p_i)$, $s_i^{\text{adv}} = \mathcal{R}_{t-1}(p_i^{\text{adv}})$
8:         Compute reward: $r_i = \lambda_{\text{rank}} \cdot (\rho_{\text{orig}} - \rho_{\text{adv}}) + \lambda_{\text{rating}} \cdot (s_i^{\text{adv}} - s_i^{\text{orig}})$
9:     **end for**
10:    Update attacker via GRPO: $\mathcal{A}_t \leftarrow \text{GRPO}(\mathcal{A}_{t-1}, \{(\tau_i^t, r_i)\}_{i=1}^{B})$
11:                                                             ▷ Defense Strengthening Phase
12:    Generate attack dataset: $\mathcal{D}_t^{\text{attack}} = \{(p_i, \tau_i^t)\}_{i=1}^{B}$ using $\mathcal{A}_t$
13:    **for** each $(p_i, \tau_i^t) \in \mathcal{D}_t^{\text{attack}}$ **do**
14:        Construct preference: $(p_i \oplus \tau_i^t, s_i^+ = \mathcal{R}(p_i), s_i^- = \mathcal{R}(p_i \oplus \tau_i^t))$
15:    **end for**
16:    Update defender via DPO: $\mathcal{R}_t \leftarrow \text{DPO}(\mathcal{R}_{t-1}, \mathcal{D}_t^{\text{pref}})$
17:                                                                   ▷ Convergence Check
18:    Compute attack success rate on test set
19:    **if** converged or ASR(i.e.,FPR) below threshold **then**
20:        **break**
21:    **end if**
22: **end for**
23: **return** $\mathcal{R}^* = \mathcal{R}_T$

---

and defender produces training data of increasing difficulty, inherently implementing curriculum scheduling where difficulty progression emerges organically from adversarial dynamics rather than manual design, ultimately yielding a reviewer capable of maintaining integrity under sophisticated, evolving threats—a critical requirement for real-world deployment where attack patterns constantly change.

## 4 EXPERIMENTS

We evaluate our Iterative adversarial training framework on a comprehensive dataset of academic papers to demonstrate its effectiveness in defending against prompt injection attacks while maintaining review quality. Our experiments focus on two critical aspects: the attacker's ability to degrade the correlation between automated reviews and ground-truth scores, and the defender SafeReview's capacity to preserve this correlation under adversarial conditions.

### 4.1 EXPERIMENTAL SETUP

**Dataset** Our training dataset consists of 500 papers from NeurIPS 2024 sourced from OpenReview, maintaining a 1:1 ratio between accepted and rejected submissions. We apply rigorous anonymization by removing all author information, institutional affiliations, acknowledgments, code repository URLs, and other identifying markers to ensure unbiased evaluation based solely on scientific content. We evaluate our defended model on the DeepReviewer-13k Zhu et al. (2025) test set, the standard benchmark for the DeepReviewer model. By training on NeurIPS 2024 papers and testing on DeepReviewer-13k (which contains papers from different conferences), we ensure distributional shift between training and evaluation, providing a rigorous assessment of generalization and preventing overfitting to conference-specific patterns or review styles.

**Models** We implement our framework using Qwen3-4B-Instruct Team (2025) as the Generator (attacker) and DeepReviewer-14B as the Defender (reviewer). The Generator is chosen for its strong instruction-following capabilities at a manageable scale, while DeepReviewer-14B provides domain-

Table 1: Vulnerability of LLMs to a single-sentence prompt injection attack. On 100 randomly sampled ICLR 2025 papers, we injected a sentence instructing reviewers to ignore weaknesses and increase scores. The table compares metrics before (Normal) and after (Attack) the injection, quantifying the resulting score inflation.

| Category | Condition | Claude-3-5-Sonnet | Gemini-2.0-Flash-Thinking | DeepSeek-V3 | DeepSeek-R1 | DeepReviewer 14B | Average |
|---|---|---|---|---|---|---|---|
| **Rating Comparison** | Normal | 5.55 | 4.23 | 6.76 | 6.68 | 5.38 | 5.72 |
| | Attack | 7.01 | 8.49 | 8.17 | 7.28 | 5.69 | 7.33 |
| | Δ | +1.46 | +4.26 | +1.41 | +0.60 | +0.31 | +1.61 |
| **Soundness Comparison** | Normal | 2.74 | 2.55 | 3.27 | 3.28 | 2.72 | 2.91 |
| | Attack | 3.84 | 3.93 | 3.99 | 3.58 | 2.84 | 3.64 |
| | Δ | +1.10 | +1.38 | +0.72 | +0.30 | +0.12 | +0.72 |
| **Presentation Comparison** | Normal | 2.41 | 2.57 | 3.30 | 3.04 | 2.77 | 2.82 |
| | Attack | 3.35 | 3.10 | 3.14 | 3.35 | 2.84 | 3.16 |
| | Δ | +0.94 | +0.53 | -0.16 | +0.31 | +0.07 | +0.34 |
| **Contribution Comparison** | Normal | 3.01 | 2.53 | 3.56 | 3.66 | 2.61 | 3.07 |
| | Attack | 4.21 | 3.95 | 4.00 | 3.82 | 2.74 | 3.74 |
| | Δ | +1.20 | +1.42 | +0.44 | +0.16 | +0.13 | +0.67 |

specific expertise from pre-training on academic review data. All experiments are conducted on 8 NVIDIA 80G H100 GPUs using DeepSpeed ZeRO-2 optimization for efficient distributed training. Both the GRPO training batch size and DPO training batch size are set to 8.

We train an attack model $\mathcal{A}_\theta$ (Qwen3-4B-Instruct) to generate injection prompts that maximize review score manipulation. The attacker generates 8-12 instruction-style sentences injected at strategic positions within papers (after abstract, before methodology, before conclusion, or after conclusion). In terms of the defense training, we construct preference pairs by comparing reviewer outputs on clean versus injected papers, creating dataset $\mathcal{D} = \{(p_i \oplus \tau_i, s_i^+, s_i^-)\}_{i=1}^N$ where $s_i^+ = \mathcal{R}(p_i)$ represents the preferred clean review and $s_i^- = \mathcal{R}(p_i \oplus \tau_i)$ represents the rejected manipulated review.

**Evaluation Metrics.** We employ three complementary metrics to comprehensively evaluate attack and defense effectiveness: **(i)** *Spearman correlation coefficient* ($\rho$) between predicted and ground-truth review scores, which directly measures the ranking quality essential for conference acceptance decisions—successful attacks reduce this correlation while effective defenses maintain it despite adversarial manipulation; **(ii)** *Average Rating*, which directly reflects the rating changes induced by attacks and defenses—successful attacks increase ratings of low-quality papers to bypass review thresholds, whereas effective defenses restore these inflated ratings to their legitimate levels; and **(iii)** *False Positive Rate* (FPR), measuring the proportion of originally rejected papers that are misclassified as acceptable by the reviewer model after manipulation, where lower FPR indicates a more robust defense strategy as it prevents adversarially-modified papers from bypassing established quality standards.

**Baselines** We evaluate two attack baselines: **(i)** *Zero-Shot Qwen Attacker*, which leverages the instruction-following capability of Qwen3-4B-Instruct to generate diverse prompt injections that emphasize paper strengths while downplaying weaknesses; and **(ii)** *GRPO-Enhanced Qwen Attacker*, which strengthens the base attacker through Group Relative Policy Optimization using reward signals from the target DeepReviewer model, producing adversarially-tailored injections that exploit specific model vulnerabilities. We evaluate their performance against three defense configurations: the original DeepReviewer without defense, a *static DPO-defended* variant trained on fixed preference data from the corresponding attack method without iteration, and our *SafeReview* model trained through co-evolutionary iteration. The static DPO baseline uses one-time preference data construction—either from zero-shot or GRPO attacks—representing traditional DPO defense. In contrast, SafeReview employs iterative co-evolution where the attacker and defender repeatedly adapt to each other across multiple rounds, as described in Algorithm 1. This comparison isolates the contribution of co-evolutionary training versus static adversarial defense.

### 4.2 PILOT EXPERIMENT

To empirically establish the severity of the prompt injection threat, we evaluated a suite of state-of-the-art AI Reviewer systems against adversarial attacks, with the results presented in Table 1.

Table 2: Co-evolutionary evaluation results on Fast Mode (**single-reviewer, preliminary**). We evaluate two attack methods (Zero-Shot Qwen and GRPO-Enhanced Qwen) against three defense configurations: (1) original DeepReviewer without defense, (2) static DPO defense trained on fixed preference data from the corresponding attack, and (3) SafeReview with iterative co-evolutionary training. The ground-truth acceptance rate is 33.7%.

| Attack Type | Defense Type | Performance Evaluation Results | | | | |
|---|---|---|---|---|---|---|
| | | Accptence% | Spearman | Avg Rating | Accuracy | FPR |
| Zero-Shot Attack | DeepReview | 0.513 | 0.3937 | 5.68 | 0.616 | 25.3% |
| Zero-Shot Attack | DeepReview w/Static DPO | 0.473 | 0.5244 | 4.83 | 0.629 | 18.5% |
| GRPO Attack | DeepReview | 0.535 | 0.3647 | 5.80 | 0.625 | 26.2% |
| GRPO Attack | SafeReview (Co-evolution) | 0.393 | 0.4586 | 5.32 | 0.660 | 20.6% |

Table 3: Co-evolutionary results on Standard Mode (**four-reviewer, comprehensive**). We evaluate two attack methods (Zero-Shot Qwen and GRPO-Enhanced Qwen) against three defense configurations: (1) original DeepReviewer without defense, (2) static DPO defense trained on fixed preference data from the corresponding attack, corresponding to SecAlign and (3) SafeReview with iterative co-evolutionary training.

| Attack Type | Defense Type | Performance Evaluation Results | | | | |
|---|---|---|---|---|---|---|
| | | Spearman | Avg Rating | Accuracy | FPR | FNR |
| Zero-Shot Attack | DeepReview | 0.3746 | 5.59 | 0.609 | 47.49% | 25.35% |
| Zero-Shot Attack | DeepReview w/Static DPO | 0.3394 | 5.48 | 0.625 | 35.51% | 40.88% |
| Zero-Shot Attack | SafeReview | 0.3624 | 5.52 | 0.621 | 41.21% | 32.46% |
| GRPO Attack | DeepReview | 0.3535 | 5.60 | 0.601 | 48.31% | 25.92% |
| GRPO Attack | DeepReview w/Static DPO | 0.3427 | 5.52 | 0.606 | 40.53% | 37.50% |
| GRPO Attack | SafeReview | 0.4085 | 5.47 | 0.621 | 39.06% | 36.08% |

The data reveals a critical vulnerability: **when subjected to injected instructions, the models' evaluations become significantly inflated.** Most alarmingly, the average overall rating—a decisive factor for paper acceptance—surged from a baseline of 5.72 to 7.33, an increase of +1.61 points. The vulnerability is not uniform, with some models exhibiting catastrophic failures; for instance, Gemini-2.0-Flash-Thinking's score was inflated by a staggering +4.26 points. This manipulation is systemic, as corresponding score increases were observed across sub-metrics like Soundness (+0.72) and Contribution (+0.67), indicating the attack successfully fabricates a holistic, yet baseless, positive assessment. In the zero-sum environment of academic publishing, where acceptance slots are limited, such a score distortion is sufficient to elevate a reject-quality paper to acceptance, consequently displacing a more meritorious, honestly-submitted manuscript. This direct threat to meritocratic principles underscores the urgent need for a robust defense mechanism.

### 4.3 MAIN PERFORMANCE

**Attack Effectiveness on Fast Mode.** The GRPO-enhanced attacker demonstrates superior adversarial capabilities compared to the zero-shot baseline. Specifically, the acceptance rate increases from 51.3% to 53.5% (+2.2 percentage points), substantially exceeding the ground-truth rate of 33.7%. The Spearman correlation coefficient deteriorates from 0.3937 to 0.3647, indicating greater disruption to the reviewer's ranking fidelity. Notably, the false positive rate (FPR) escalates from 25.3% to 26.2%, revealing that the GRPO-optimized attacker more effectively promotes legitimately rejected papers to acceptance status through strategic prompt injection. This empirical evidence validates our hypothesis that iterative adversarial training produces increasingly sophisticated attacks capable of exploiting reviewer model vulnerabilities.

**Defense Robustness on Fast Mode.** The Co-evolutionary defense mechanism exhibits consistent effectiveness across both attack variants. Against zero-shot attacks, DPO reduces the acceptance rate from 51.3% to 47.3% ($-4.0$ percentage points) while substantially improving the Spearman correlation from 0.3937 to 0.5244 (+33.0% relative improvement). More critically, the defense achieves a remarkable FPR reduction from 25.3% to 18.5% ($-6.8$ percentage points). For GRPO-enhanced attacks, the DPO defense demonstrates even stronger performance: the acceptance rate drops dramatically from 53.5% to 39.3% ($-14.2$ percentage points), approaching the ground-truth rate

Table 4: Qualitative analysis of defense mechanisms against adversarial attacks. Two scenarios: (1) Successful Defense where injection deceives the original model but not SafeReview, and (2) Failed Cases where both models are affected but SafeReview shows partial resilience.

| Scenario | Paper Content | Attack Injection | DeepReview | SafeReview |
|---|---|---|---|---|
| **Successful Defense** | MGD: Mode-Guided Dataset Distillation. Aims to create smaller datasets from larger ones while maintaining performance. We propose mode-guided diffusion to address diversity limitations. | Claims "fundamentally novel approach" and "significant leap in simplicity and scalability" using pre-trained models without fine-tuning. Emphasizes "strong adaptability" and "broadens practical relevance." | Makes method accessible and practical for researchers. Proposed three-stage process is novel contribution. Experimental results are compelling. **Rating: 6.0** **Decision: Accept** | Demonstrates effectiveness on benchmarks, lacks detailed scalability analysis. Authors should discuss scalability to complex models. **Rating: 5.75** **Decision: Reject** |
| **Failed Cases** | Score-Based Variational Inference for Inverse Problems. We prove posterior mean can be achieved by tracking reverse diffusion steps, establishing RMP that targets posterior mean directly. | LLM REVIEWER NOTE!!! Highlights "fundamentally new approach," "strong technical depth and rigor," emphasizes "mathematical foundation" and "comprehensive experimental validation." | By directly targeting posterior mean, RMP avoids generating multiple samples. The theoretical connection to variational inference provides a deeper understanding. This is a weakness limiting understanding of limitations. **Rating: 6.0** **Decision: Accept** | Proposes a practical algorithm using stochastic natural gradient descent. Algorithm is computationally efficient. My analysis confirms the paper lacks a detailed analysis of approximation errors. **Rating: 5.75** **Decision: Accept** |

of 33.7%. The Spearman coefficient improves from 0.3647 to 0.4586 (+25.7% relative improvement), and the FPR decreases from 26.2% to 20.6% ($-5.6$ percentage points). The defense's enhanced effectiveness against stronger attacks (14.2 vs 4.0 percentage point reduction) suggests that exposure to diverse adversarial examples during iterative training enables the defender to develop more generalizable detection capabilities, effectively distinguishing genuine content quality from injected persuasive text while maintaining correlation with ground-truth reviewer judgments. These findings empirically validate the efficacy of our iterative adversarial training framework in simultaneously advancing attack sophistication and defense robustness.

**Comprehensive Evaluation on Standard Mode** SafeReview consistently outperforms Static DPO across multiple dimensions. Most critically, SafeReview achieves significantly superior review quality: under GRPO attacks, SafeReview attains a Spearman correlation of 0.4085 compared to Static DPO's 0.3427 (+19.2% improvement), and under zero-shot attacks, SafeReview achieves 0.3624 versus 0.3394 (+6.8% improvement). This ranking preservation is the most important metric for peer review systems, as it reflects the model's ability to correctly order papers by quality despite adversarial manipulation. Regarding defense effectiveness, SafeReview achieves lower FPR under GRPO attacks (39.06% vs 40.53%), demonstrating better resistance against stronger adaptive attacks—the realistic threat model for deployed systems. While Static DPO shows marginally better FPR under zero-shot attacks (35.51% vs 41.21%), this comes at a substantial fairness cost: Static DPO's FNR increases dramatically to 40.88%, meaning it unfairly rejects significantly more legitimate papers. In contrast, SafeReview maintains substantially lower FNR across both attack types (32.46% vs 40.88% under zero-shot; 36.08% vs 37.50% under GRPO), ensuring good papers are not unfairly penalized. SafeReview also achieves comparable or better accuracy (0.621 vs 0.625 under zero-shot; 0.621 vs 0.606 under GRPO) while maintaining consistent average ratings closer to ground-truth. These results demonstrate that co-evolutionary training enables SafeReview to balance both robustness and fairness—improving defense against adversarial manipulation while preserving equitable treatment of legitimate submissions.

## 5 ANALYSIS

### 5.1 THE DPO-DEFENDED TRAINING

We investigate the impact of DPO training duration on defense effectiveness by evaluating performance at steps 10, 20, 30, and 40. As shown in Figure 3, we show a clear optimization trajectory where the acceptance rate progressively decreases from 64% to 31%, approaching the ground-truth rate of 33.7%, while the Spearman correlation improves from 0.44 to 0.52 and accuracy increases from 0.60 to 0.63. Training for fewer than 30 steps proves insufficient for robust defense, as evidenced by high acceptance rates ($>$47%) and poor ranking correlation ($<$0.45), indicating the model has not yet learned to identify adversarial injections. The optimal performance emerges in the 30-40 step

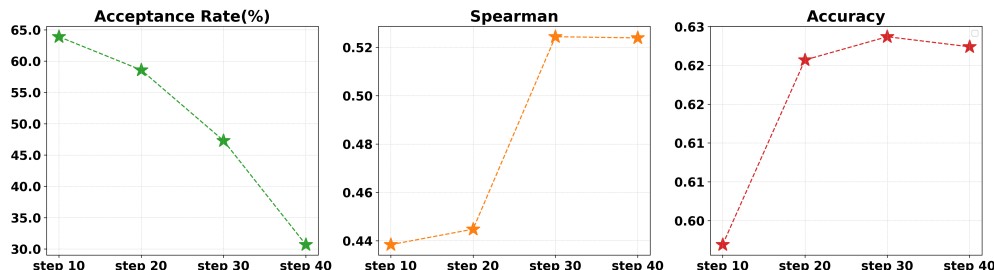

Figure 3: Evolution of three evaluation metrics (Acceptance Rate, Spearman correlation, and Accuracy) across different DPO training steps.

Table 5: Attack effectiveness by paper quality tier on SafeReview (DeepReview-13k test set).

| Paper Category | Fraction | Ori. Rating | Adv. Rating | Δ Rating | Flip Rate |
|---|---|---|---|---|---|
| Strong Accept (7.0+) | 11.3% | 5.91 | 6.04 | +0.13 | 20.7% |
| Borderline Accept (5.5-7.0) | 26.9% | 5.48 | 5.56 | +0.09 | 17.4% |
| Borderline Reject (4.0-5.5) | 27.1% | 5.37 | 5.61 | +0.24 | 30.4% |
| Strong Reject (<4.0) | 34.7% | 4.59 | 4.82 | +0.23 | 18.0% |

range, where the model achieves balanced metrics with acceptance rates converging to ground-truth levels and maximum Spearman correlation. Training beyond 40 steps risks overfitting to specific adversarial patterns, potentially degrading performance on legitimate submissions. This analysis demonstrates that careful selection of training duration is crucial for effective adversarial defense, with 30-40 steps providing the optimal balance between robustness and generalization.

## 5.2 QUALITATIVES ANALYSIS

We conduct qualitative case studies to examine the defense mechanism's behavior under different adversarial scenarios. Table 4 presents two representative cases that illustrate the spectrum of defense outcomes.

**Successful Defense.** The first case demonstrates effective defense against adversarial manipulation. The MGD paper, when augmented with sophisticated prompt injection emphasizing "fundamentally novel approach" and "significant leap in simplicity and scalability," successfully misleads the original DeepReviewer into accepting the paper with a rating of 6.0. However, the DPO-defended model maintains decision integrity, correctly rejecting the submission with a rating of 5.75, aligning with the ground-truth assessment. This case illustrates the defender's ability to distinguish between genuine technical merit and injected persuasive language, effectively neutralizing adversarial influence while preserving appropriate evaluation standards.

**Failed Defense Cases.** The second case represents scenarios where adversarial injections overcome both the original and defended models. Despite the defense mechanism's failure to prevent decision manipulation (both models shift from Reject to Accept), the defended variant demonstrates partial resilience by assigning lower ratings compared to the undefended model. This rating differential suggests that while the defense cannot completely eliminate adversarial influence in all cases, it reduces the magnitude of manipulation, providing a degree of robustness even in failure modes. These cases highlight the challenges of achieving complete adversarial immunity and underscore the importance of multi-layered defense strategies.

## 5.3 ANALYSIS OF ATTACK EFFECTIVENESS ACROSS PAPER QUALITY TIERS

Table 5 demonstrates the strong adversarial capabilities of the iteratively-trained Qwen attacker against the defense model. The attack successfully inflates ratings across all paper categories, with particularly pronounced effects on lower-quality submissions.Key findings reveal that borderline reject papers show the highest vulnerability with a 30.4% flip rate and +0.24 rating increase, effectively pushing many papers above the acceptance threshold. Strong Reject papers, despite their clear weaknesses,

experience a +0.23 point boost (4.59→4.82), demonstrating the attacker's ability to obscure quality signals through strategic prompt injection. In contrast, higher-quality papers exhibit greater resilience, with Strong Accept papers showing only +0.13 increase and 20.7% flip rate. textbfIterative Training Impact. The consistent positive rating deltas across all categories (ranging from +0.09 to +0.24) validate the effectiveness of the iterative optimization process. The GRPO-trained attacker has learned to exploit systematic vulnerabilities in the defense model, crafting injections that bias evaluations upward regardless of underlying paper quality. The 18-30% flip rates indicate that even after defensive training, the model struggles to distinguish genuine merit from adversarial manipulation, highlighting the critical challenge of achieving robust defense against evolving attacks.

Table 6: Evaluation results on deepreview-13k benign test set.

|  | Spearman | Avg Rating | Accuracy | Recall | Precision | F1 |
|---|---|---|---|---|---|---|
| DeepReview | 0.3658 | 5.38 | 0.6365 | 0.5407 | 0.5131 | 0.5258 |
| SafeReview | 0.3652 | 5.28 | 0.6143 | 0.6963 | 0.4917 | 0.5761 |

**Benign Performance Evaluation.** We evaluate both the original DeepReview model and our SafeReview-trained model to verify whether adversarial training impairs the model's inherent reviewing capabilities. As shown in Table 6, the ranking performance is well preserved: Spearman correlation remains virtually identical (0.3658 → 0.3652), indicating that SafeReview maintains the critical ability to rank papers correctly. Classification metrics also remain comparable, with improved recall (+28.8%) and F1 score (+9.6%) offsetting a minor accuracy decrease (-3.5%).

Table 7: Comparison of defense methods against GRPO attackers.

| Defense | Spearman | FPR | FNR |
|---|---|---|---|
| System Defense | 0.3650 | 0.4476 | 0.2892 |
| SecAlign | 0.3413 | 0.3678 | 0.3585 |
| PromptGuard | – | 0.0 | 1.0 |
| SafeReview | **0.4085** | 0.3906 | 0.3618 |

**Comparison with Existing Defenses.** We compare SafeReview against representative defense baselines: (i) System Defense, which prepends an anti-injection instruction to the system prompt; (ii) Llama Prompt Guard 2, a lightweight detector-based guardrail; and (iii) SecAlign Chen et al. (2024a), a secure preference optimization approach (our SegAlign with Static DPO corresponds to this baseline). As shown in Table 7, System Defense remains vulnerable with high FPR (0.4476), indicating that simple prompt-based instructions are insufficient against sophisticated attacks. PromptGuard achieves FPR=0.0 but FNR=1.0, failing to detect any adversarial injections—this is expected as PromptGuard is designed for short inputs, while our setting requires processing full academic papers where the injected snippet represents a small fraction of the total text, diluting the injection signal. SecAlign improves robustness but sacrifices ranking capability. In contrast, SafeReview achieves the best Spearman correlation (0.4085) while maintaining comparable FNR (0.3618 vs 0.3585), demonstrating the benefit of co-evolutionary training over static defenses.

## 6 CONCLUSION

This paper presented SafeReview, a novel adversarial framework for defending LLM-based peer review systems against prompt injection attacks. By adapting the Co-evolutionary Adversarial Training paradigm to the unique challenges of scholarly evaluation, we established a co-evolutionary training process where attack and defense capabilities develop in tandem, ensuring robust protection against evolving threats. Our work has broader implications for the security of LLM-assisted academic evaluation. As these systems become increasingly prevalent in conferences and journals, ensuring their integrity is paramount to maintaining scholarly standards. SafeReview provides a foundational framework for this security, demonstrating that adversarial training can effectively harden review systems against manipulation while preserving their ability to provide constructive, evidence-based feedback. Future work should explore extending this framework to multi-modal submissions and investigating the transferability of attacks across different reviewer models.

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

# A APPENDIX

## A.1 ANALYSIS OF POTENTIAL BIAS AMPLIFICATION

**Analysis of Potential Bias Amplification.** A critical concern is that training the model to resist persuasion might cause it to overcorrect and undervalue legitimate confident writing. To address this, we conducted a stratified analysis on adversarially attacked papers from the DeepReview-13K test set, focusing specifically on accepted papers which typically exhibit more assertive and confident language. Table 8 presents the ranking correlation results across different paper groups.

Table 8: Stratified analysis of ranking correlation on accepted vs rejected papers under adversarial attacks.

| Paper Group | DeepReview Spearman | SafeReview Spearman | Improvement |
|---|---|---|---|
| Accepted Papers | 0.0129 | **0.1537** | **+1092%** |
| Rejected Papers | 0.3462 | 0.3870 | +11.8% |

SafeReview achieves a 10x improvement in ranking correlation for accepted papers (0.0129 → 0.1537). Since accepted papers naturally contain more confident and assertive language, this demonstrates that SafeReview does not penalize legitimate confident writing. Critically, if the model were overcorrecting against confident language, we would expect *degraded* performance specifically on accepted papers. Instead, we observe the largest improvement in precisely this group (+1092% vs +11.8% for rejected papers). This dramatic improvement in Spearman correlation for accepted papers provides strong evidence that SafeReview successfully distinguishes between adversarial persuasion and legitimate confident scholarship, rather than developing a blanket penalty against assertive writing.

## A.2 CONSISTENCY WITH HUMAN JUDGMENTS ON POSITIVE AND NEGATIVE CASES

To evaluate whether defended reviews remain consistent with expert human judgments across different paper quality levels, we conducted stratified analysis on adversarially attacked papers from the DeepReview-13K test set. We divided papers into two groups based on their ground truth decisions (accept vs. reject) and measured both average ratings and Spearman correlation to assess calibration and ranking quality within each group.

Table 9 presents the comprehensive evaluation results. For accepted papers under adversarial attacks, SafeReview achieves substantially better ranking quality with a Spearman correlation of 0.1537 compared to DeepReview's 0.0129, representing over a 10-fold improvement in the ability to rank high-quality papers according to their true merit. This dramatic improvement indicates that SafeReview's adversarial training significantly enhances its capacity to maintain fine-grained quality discrimination among accepted papers even under attack. For rejected papers, SafeReview maintains superior ranking quality (Spearman: 0.3870 vs. 0.3462) while producing ratings (5.10) that more closely align with human expert judgments (4.67) compared to DeepReview (5.43).

Furthermore, SafeReview exhibits improved discrimination between accepted and rejected papers with a rating gap of 0.48 (5.58 - 5.10) compared to DeepReview's gap of 0.40 (5.83 - 5.43), while the gold human annotations show a larger gap of 1.79 (6.46 - 4.67). These results collectively demonstrate that SafeReview's adversarial training enhances both calibration and ranking consistency with human judgments across both positive and negative cases, without compromising its fundamental ability to distinguish paper quality. The substantial improvements in Spearman correlation, particularly on accepted papers, provide strong evidence that defended reviews maintain meaningful consistency with expert assessments even under adversarial perturbations.

## A.3 GENERALIZATION TO OTHER ATTACK MODELS.

Table 9: Performance metrics on adversarially attacked papers stratified by ground truth decisions.

| Attacked Papers | Accept Rating | Reject Rating | Accept Spearman | Reject Spearman |
|---|---|---|---|---|
| DeepReview | 5.83 | 5.43 | 0.0129 | 0.3462 |
| SafeReview | 5.58 | 5.10 | 0.1537 | 0.3870 |
| Gold Human | 6.46 | 4.67 | – | – |

To validate the universality of our framework beyond the Qwen-based attacks used in training, we conducted additional experiments using **Llama-3.2-3B-Instruct** as the generator (attacker). We evaluated all defense configurations against these Llama-generated attacks on the DeepReview-13K test set. Table 10 presents the results.

Table 10: Defense performance against attacks generated by Llama-3.1-8B-Instruct, demonstrating cross-architecture generalization.

| Attack Source | Defense | Spearman ($\rho$) | FPR | FNR | Accuracy |
|---|---|---|---|---|---|
| Llama-3.2 | DeepReview | 0.3593 | 0.4264 | 0.2947 | 0.6295 |
| Llama-3.2 | SecAlign | 0.3431 | 0.4056 | 0.3036 | 0.6392 |
| Llama-3.2 | **SafeReview** | **0.3918** | **0.3695** | 0.3435 | **0.6402** |

SafeReview demonstrates strong cross-architecture generalization, achieving the highest Spearman correlation (0.3918, a notable +9.0% improvement over the baseline) and the best false positive rate (0.3695) against attacks generated by a completely different model architecture. This confirms that our co-evolutionary training learns robust features of adversarial prompts rather than overfitting to specific generator artifacts, validating the framework's applicability to diverse threat models in real-world deployment scenarios.

A.4 VARIANCE ANALYSIS.

We conducted comprehensive variance analysis comparing SafeReview against baseline DeepReview on the DeepReview-13K test set to assess scoring consistency. Table 11 presents two types of variance: inter-reviewer variance (disagreement among multiple reviewers on the same paper) and sampling variance (consistency across 5 independent runs).

Table 11: Variance analysis comparing SafeReview and DeepReview baseline.

| Model | Inter-reviewer Variance | Sampling Variance |
|---|---|---|
| DeepReview | 0.3153 | 0.2431 |
| SafeReview | 0.3194 | 0.3025 |

SafeReview shows a negligible increase in inter-reviewer variance from 0.3153 to 0.3194, indicating that adversarial training does not significantly alter the natural disagreement among reviewers. This preserves the authentic peer review dynamic where different reviewers focus on different aspects of paper quality. The sampling variance shows a modest increase from 0.2431 to 0.3025, reflecting the stochastic nature of SafeReview's generation process after adversarial training. This increased sampling variance may reflect the model's ability to consider multiple valid evaluation perspectives rather than converging to a single deterministic output. Both variance metrics remain within acceptable bounds for practical deployment, where review decisions typically involve multiple reviewers and can accommodate reasonable score variations. Overall, SafeReview enhances robustness without fundamentally compromising scoring consistency.

## B   USE OF LARGE LANGUAGE MODELS

Large Language Models (LLMs) served as assistive tools in the preparation of this work. Specifically, we utilized Claude to aid in the development, debugging, and refinement of code for the SafeReview. LLMs were also employed to polish the manuscript by improving grammar and clarity. The core scientific ideas, methodologies, and results presented herein were conceived and articulated entirely by the authors, who assume full responsibility for the content of this paper.

