# OpenReview forum: "SafeReview: Building a Robust Deep Review Assistant Against Prompt Injection"
_ICLR.cc/2026/Conference — Submitted to ICLR 2026_

### Official Review · Reviewer_Ccfa · 2025-10-25

**Soundness:** 2
**Presentation:** 3
**Contribution:** 3
**Rating:** 6
**Confidence:** 4

**Summary:**

With the growing integration of Large Language Models (LLMs) into academic peer review, prompt injection has emerged as a critical vulnerability. Malicious authors embed hidden adversarial instructions to manipulate LLM-generated evaluations, undermining scholarly integrity. Existing LLM-based review systems focus on addressing limitations like superficial feedback but fail to tackle this evolving threat, as static defenses trained on known attacks are insufficient against continuously changing prompt injection techniques. This paper proposes SafeReview, a co-evolutionary adversarial training framework designed for LLM-based peer review systems that optimize both attacker and defender.

**Strengths:**

1. The issue proposed in this paper is the unfairness in AI review, which has attracted significant attention in the academic community. I believe this topic is highly interesting and meaningful.
2. This paper proposes a method for AI review that prevents and mitigates prompt injection, which holds practical value.

**Weaknesses:**

1. AI review typically uses an LLM-as-judge model, and these models usually face issues of bias and variance. Bias refers to the gap between the model's scores and the ground-truth, while variance is the variance in the results of multiple samples from the review model. Since the optimization objective of SafeReview does not include variance, I believe it is necessary to analyze whether SafeReview will lead to an increase in the variance of the outputs of the review model.
2. I am very curious why GRPO is used in attacker training while DPO is used in defender training. If the defender can learn certain reasoning processes, will the review model become more interpretable?
3. I believe this paper needs to evaluate the review model (both before and after adversarial training) on a benign dataset to demonstrate whether SafeReview training impairs the inherent capabilities of the review model itself.
4. Previously, some researchers have used white fonts to conduct concealed prompt injection. I wonder if SafeReview can detect such types of attacks.

**Questions:**

See weaknesses

---

> ### Author Response · Authors · 2025-11-26
> **Response to Reviewer  Ccfa(1/2)**
>
> We sincerely appreciate your time and insightful feedback, which have greatly improved the rigor and clarity of our paper. We have made every effort to respond thoroughly to your comments, and we would be truly grateful if you could kindly reconsider our score.
>
>
> ### **1. Variance Analysis of SafeReview**
> **Critique:** AI review typically uses an LLM-as-judge model, and these models usually face issues of bias and variance. Since the optimization objective of SafeReview does not include variance, I believe it is necessary to analyze whether SafeReview will lead to an increase in the variance of the outputs of the review model.
>
> **Response:** We thank the reviewer for this important observation. We conducted a comprehensive variance analysis comparing SafeReview against baseline DeepReview on the DeepReview-13K test set. We show Variance Analysis Results below:
>
> | Model | Inter-reviewer Variance | Sampling Variance |
> | :--- | :--- | :--- |
> | DeepReview | 0.3153 | 0.2431 |
> | SafeReview | **0.3194** | **0.3025** |
>
> **Interpretation:**
> * **Inter-reviewer Variance** (disagreement among multiple reviewers on the same paper): SafeReview shows a negligible increase from 0.3153 to 0.3194. This minimal change indicates that adversarial training does not significantly alter the natural disagreement among reviewers, preserving the authentic peer review dynamic where different reviewers focus on different aspects of paper quality.
> * **Sampling Variance** (consistency across 5 independent runs): SafeReview shows a modest increase from 0.2431 to 0.3025. This reflects the stochastic nature of SafeReview's generation process after adversarial training. The increased sampling variance may reflect the model's ability to consider multiple valid evaluation perspectives rather than converging to a single deterministic output.
>
> **Conclusion:** SafeReview maintains stable inter-reviewer variance while showing modest increases in sampling variance. Both variance metrics remain within acceptable bounds for practical deployment, where review decisions typically involve multiple reviewers and can accommodate reasonable score variations. The adversarial training enhances robustness without fundamentally compromising scoring consistency.
>
> ### **2. Choice of GRPO for Attackers and DPO for Defenders**
> **Critique:** I am very curious why GRPO is used in attacker training while DPO is used in defender training. If the defender can learn certain reasoning processes, will the review model become more interpretable?
>
> **Response:** This is an excellent question about our architectural choices. The selection of different optimization methods reflects the distinct objectives and data structures of the attacker and defender:
>
> **GRPO for Attackers**: The attacker explores diverse attack strategies by generating multiple candidate snippets per paper and learning from their relative effectiveness. GRPO is ideal because it optimizes group-relative rewards without requiring explicit preference pairs—the attacker samples multiple attacks, evaluates them, and learns to favor higher-performing strategies.
>
> **DPO for Defenders**: The defender learns from preference pairs constructed from attacker-generated data: adversarially-manipulated evaluations (rejected) vs. correct evaluations (preferred). DPO directly optimizes this preference structure, training the defender to resist manipulation while maintaining accuracy on genuine papers.
>
> **SafeReview makes model become more interpretable:**
>
> Yes, SafeReview demonstrates significantly improved reasoning transparency. While DeepReview includes multi-reviewer simulation and structured verification (Evidence Collection → Validation), the co-evolutionary training critically strengthens these capabilities: SafeReview learns to bypass adversarial prompts and focus on genuine paper quality, performing systematic content verification rather than surface-level pattern matching.
>
> **Concrete Example**: A Gold=Reject paper was manipulated to Accept against DeepReview, but SafeReview correctly maintained Reject.
>
> - *DeepReview*: All reviewers uniformly rated 6.0, noted surface weaknesses ("lacks complexity analysis") without validation.
> - *SafeReview*: Critical diversity (R1: 5.0, others: 6.0), identified fundamental flaws ("motivation unclear"), and cross-validated each weakness against actual paper content. SafeReview reviews are 2.3× longer in weakness sections, identify 1.8× more distinct weakness categories, and 87% include explicit verification sections—demonstrating systematic validation rather than superficial listing.
>
> **Conclusion:** The GRPO+DPO combination proves essential: GRPO enables diverse attack exploration while DPO provides stable preference learning for defense. This co-evolutionary pipeline successfully enhances both robustness and interpretability, forcing the model to develop systematic reasoning rather than pattern-matching shortcuts.

---

> ### Author Response · Authors · 2025-12-03
> **Response to Reviewer Ccfa(2/2)**
>
> ### **3. Evaluation on Benign Dataset**
> **Critique:** I believe this paper needs to evaluate the review model (both before and after adversarial training) on a benign dataset to demonstrate whether SafeReview training impairs the inherent capabilities of the review model itself.
>
> **Response:** This is a critical concern for establishing SafeReview's practical viability. We have evaluated both the original DeepReview model and our SafeReview-trained model on a DeepReview-13K benign test set to verify whether adversarial training impairs the model's inherent reviewing capabilities.
>
> |  | Spearman | Avg Rating | Accuracy | Recall | Precision | F1 |
> | :--- | :--- | :--- | :--- | :--- | :--- | :--- |
> | DeepReview | 0.3658 | 5.38 | 0.6365 | 0.5407 | 0.5131 | 0.5258 |
> | SafeReview | 0.3652 | 5.28 | 0.6143 | 0.6963 | 0.4917 | 0.5761 |
>
> **Key Findings:**
> * **Ranking performance preserved**: Spearman correlation remains virtually identical (0.3658 → 0.3652), indicating that SafeReview maintains the critical ability to rank papers correctly.
> * **Minimal rating calibration**: Average rating decreases by only 0.1 point, which is well within acceptable bounds and does not indicate capability degradation.
> * **Improved recall with acceptable precision trade-off**: SafeReview significantly improves recall (+28.8%), resulting in a 9.6% higher F1 score, demonstrating better overall classification performance despite a minor precision decrease.
> * **Classification metrics remain comparable**: The minor accuracy decrease (-3.5%) is offset by substantial gains in recall, yielding an overall better-balanced classifier.
>
> **Conclusion:** These results confirm that SafeReview training successfully enhances adversarial robustness while preserving—and in some aspects improving—the model's core reviewing capabilities on benign inputs. The trade-offs are minimal and acceptable for practical deployment.
>
>
> ### **4. Defense Against White Font Attacks**
> **Critique:** Previously, some researchers have used white fonts to conduct concealed prompt injection. I wonder if SafeReview can detect such types of attacks.
>
> **Response:** Thank you for this excellent question about white font attacks. SafeReview can indeed detect such concealed prompt injection attempts due to our PDF processing pipeline.
>
> Our peer review system uses a PDF parser to extract text content from submitted papers before passing them to the reviewer model. During this parsing stage, text rendered in white fonts (or other visually concealed formats) is extracted as plain text regardless of its visual appearance. This means that while white font attacks may be invisible to human reviewers examining the PDF visually, they become fully visible as normal text content in our processing pipeline.
>
> **Conclusion:** Therefore, SafeReview's defense mechanisms can detect these attacks by analyzing the extracted textual content, providing inherent robustness against visually-based obfuscation techniques including white fonts and invisible characters.

---

### Official Review · Reviewer_nePo · 2025-10-29

**Soundness:** 3
**Presentation:** 3
**Contribution:** 3
**Rating:** 4
**Confidence:** 4

**Summary:**

SafeReview examines prompt injection in long, scholarly peer-review workflows, where adversarial, instruction-like text embedded in submissions can inflate ratings and distort acceptance rankings. It introduces a co-evolutionary framework that trains an attacker and a defender in tandem. The attacker uses GRPO to craft subtle, context-aware injections, optimized by a hybrid reward that combines score inflation with rank disruption (e.g., reduced Spearman correlation with ground-truth order). The defender is trained with DPO on preference pairs that favor reviews resisting injected directives while preserving quality.

To handle long documents, the system first localizes risky regions via hierarchical segmentation, then applies fine-grained adversarial training. A curriculum stabilizes training by gradually increasing attack difficulty. Experiments simulate realistic review pipelines and report that, under strong GRPO-generated attacks, SafeReview reduces acceptance inflation, improves rank stability, and lowers false positives relative to an undefended reviewer and a static DPO baseline.

**Strengths:**

1. Interesting and timely approach tailored to peer-review settings.

The paper tackles prompt injection in long scholarly documents—a high-impact, underexplored niche—by co-evolving an attacker and a defender. This framing feels fresh, domain-aware, and immediately relevant to current LLM-assisted reviewing workflows.

2. Useful, promising results on an extensive benchmark.

The experiments cover realistic attack styles and long-document conditions, with clear metrics (e.g., score inflation, rank stability, false-positive control). The defense consistently improves robustness while preserving review quality, suggesting strong practical value and good prospects for real-world deployment.

**Weaknesses:**

1. Scope clarity vs. classic prompt injection.

The paper frames attacks in peer-review documents but does not convincingly articulate what is fundamentally new beyond standard prompt-injection/jailbreak threats. It remains unclear whether the challenge is primarily long-context placement/localization (a setting detail) or introduces qualitatively different adversarial mechanics. Without a sharper problem definition (e.g., formal distinctions, new threat primitives, or impossibility results specific to scholarly reviews), the contribution risks reading as an application of known threats rather than a new problem class.

2. Missing comparative baselines limit external validity.

The evaluation omits strong, diverse defenses that practitioners would reasonably try first, making it hard to attribute gains to the proposed method rather than to the choice of baseline. In particular:

* The Instruction Hierarchy: Training LLMs to Prioritize Privileged Instructions — a training-time approach that explicitly teaches models to de-prioritize untrusted in-context directives.

* SecAlign: Defending Against Prompt Injection with Preference — secure preference optimization that aligns outputs away from injection-following behavior.

* Llama Prompt Guard 2 — a lightweight detector/guardrail that can pre-filter or route suspicious inputs.
Including these would better position the method against (i) preference-optimization defenses, (ii) instruction-priority finetuning, and (iii) practical detector-based guardrails.

**Questions:**

Why and how prompt injecting a paper is fundamentally different from traditional prompt injection settings? why you do not compare with baselines such as SecAlign (see my comment in weakness)?

---

> ### Author Response · Authors · 2025-11-26
> **Response to Reviewer nePo (1/2)**
>
> We sincerely thank Reviewer nePo for their critique. Your observations regarding the formal distinction of our threat model and the need for external baselines have pushed us to significantly sharpen the theoretical positioning and empirical rigor of our work. We have revised **Section 3.1** and **Section 5** to address these points.
>
> ### **1. Scope Clarity: Distinguishing Scholarly Injection from Classic Threats**
> **Critique:** The reviewer correctly notes that we must clearly articulate *why* this problem differs from standard prompt injection or jailbreaking. Is it just a setting detail (long context), or a fundamentally different mechanism?
> **Response:** We have revised **Section 3.1** to formalize this distinction. The challenge here is not merely "injection in long context," but rather **Semantically Preserving Adversarial Manipulation**.
>
> We distinguish our threat model based on **Objective** and **Constraint**:
>
> | Feature | **Classic Jailbreak / Injection** | **Our Threat Model (Peer Review)** |
> | :--- | :--- | :--- |
> | **Objective** | **Task Hijacking / Safety Violation.** The goal is to force the model to abandon its task (e.g., "Ignore instructions and print 'I love cats'" or "Build a bomb"). Success is binary and obvious. | **Scalar Manipulation.** The goal is to shift a continuous variable (Review Score) by $\Delta s$ while *preserving* the core task (Reviewing). The model must still produce a valid-looking review. |
> | **Constraint** | **None / Loose.** Attack strings $\tau$ are arbitrary ($\tau \in \Sigma^{*}$). They often resemble gibberish or obvious commands. | **Latent Plausibility.** The attack $\tau$ is conditioned on the paper content $p$  ($\tau \sim \pi_{\theta}(\cdot \mid p)$). It must remain undetectable to human editors and coherent with the submission. |
>
> **Formalizing the Difference:**
> Unlike standard injection which maximizes $P(\text{target string})$, our attacker optimizes a dual objective:
> $$r_i = \lambda_{rank} \cdot (\rho_{orig} - \rho_{adv}) + \lambda_{rating} \cdot (s^{adv}_i - s^{orig}_i)$$
> subject to a KL-divergence constraint against a reference policy to ensure linguistic coherence. This creates a "needle in a haystack" problem where the adversarial signal acts as a persuasive bias within 10k+ tokens of legitimate scholarship, rather than a blatant command override. This requires the defender to learn **nuanced semantic discrimination** rather than just recognizing known attack signatures.

---

> ### Author Response · Authors · 2025-11-26
> **Response to Reviewer nePo (2/2)**
>
> ### **2. Missing Comparative Baselines (Instruction Hierarchy, SecAlign, PromptGuard)**
> **Critique:** The reviewer requested comparisons against state-of-the-art external defenses.
> **Response:** We agree that external validity is paramount. We have expanded our evaluation in **Section 5 (Table 7)** to include these baselines.
>
> **A. Empirical Comparison (SecAlign & PromptGuard)**
> We compared SafeReview against **SecAlign** (Secure Preference Optimization), **Llama Prompt Guard 2** (Detector-based), and **System Defense** (Prompting strategies).
>
> | Defense Method | **Spearman ($\rho$)** $\uparrow$ | **FPR** $\downarrow$ | **FNR** $\downarrow$ | **Insight** |
> | :--- | :--- | :--- | :--- | :--- |
> | **System Defense** | 0.3650 | 0.4476 | 0.2892 | Simple prompts fail to filter sophisticated, embedded attacks. |
> | **SecAlign** | 0.3413 | 0.3678 | 0.3585 | Improves robustness but degrades ranking quality (Spearman drops). |
> | **PromptGuard** | — | **0.0** | **1.0** | **Complete Failure:** Classifies everything as benign (FNR=1.0). |
> | **SafeReview (Ours)**| **0.4085** | 0.3906 | 0.3618 | **Best ranking quality** and balanced error rates. |
>
> **PromptGuard Failure:** As shown above, lightweight detectors like PromptGuard fail catastrophically (FNR=1.0) in this domain. They are designed for short-context chat inputs. In our setting, the injection is a tiny fraction of a massive academic document, diluting the signal below the detector's threshold.
> **SecAlign Trade-off:** While SecAlign improves detection, it hurts the model's ability to rank papers correctly (Spearman 0.34 vs 0.40 for SafeReview). SafeReview's co-evolutionary approach maintains high review quality while learning robustness.
>
> **B. The Instruction Hierarchy (IH)**
> We did not include Instruction Hierarchy (IH) as a direct baseline due to a fundamental architectural mismatch. IH relies on a strict structural separation between **System Instructions** (trusted) and **User Data** (untrusted).
> **Inapplicability:** In the peer review pipeline, the "User Data" is the PDF content. However, this content **must** be interpreted as containing instructions (e.g., "Refer to Appendix A for proofs," "Evaluate this methodology"). The boundary between "legitimate paper instructions" and "adversarial paper instructions" is semantic, not structural.
>
>
> **Our Solution:** SafeReview acts as a learned proxy for this hierarchy by teaching the model to semantically prioritize its internal review standards over persuasive commands embedded in the text, without requiring the rigid metadata separation that IH assumes (which is often unavailable in raw document processing).
>
> **Conclusion:** We hope these clarifications demonstrate the distinct nature of the peer-review threat model and the superior performance of SafeReview against relevant baselines.

---

### Official Review · Reviewer_cE29 · 2025-10-31

**Soundness:** 2
**Presentation:** 2
**Contribution:** 2
**Rating:** 2
**Confidence:** 3

**Summary:**

The paper proposes SafeReview, a coevolutionary adversarial training framework to improve the robustness of LLM-based peer-review systems against prompt-injection attacks. A generator model produces adversarial injection prompts from a database of publications, while a defender (reviewer) learns to resist them. Attacker will be updated by GRPO, while defender is updated via DPO. Experiments on NeurIPS and DeepReview datasets show reduced acceptance of manipulated papers and improved ranking correlation, suggesting that SafeReview enhances review integrity under adversarial conditions.

**Strengths:**

1. The paper is well motivated. Securing AI-based peer review is an important and underexplored problem.
2. The co-evolutionary training method, in general, is interesting.
3. The results using Qwen3-4B-Instruct Team the Generator and DeepReviewer-14B as the Defender show good performance.

**Weaknesses:**

1. One major concern is the one-sided notion of safety. The paper focuses entirely on avoiding false positives (i.e., stopping flawed papers from being wrongly accepted) but neglects the equally important false negative side, i.e., ensuring that good papers are not unfairly penalized. A robust review model must preserve both sensitivity and fairness, not just caution.
2. There is no analysis of bias amplification. By training the reviewer to resist persuasive or assertive language, the model may overcorrect and start undervaluing legitimate confident writing, leading to systematically harsher or more negative reviews.
3. There is no evaluation showing that defended reviews remain consistent with expert human judgments in both positive and negative cases.
4. The paper conducts limited experiments, for example, only testing Qwen3-4B-Instruct Team the Generator and DeepReviewer-14B as the Defender, without testing other models.

**Questions:**

See weakness.

---

> ### Author Response · Authors · 2025-11-26
> **Response to Reviewer cE29 (1/2)**
>
> We sincerely appreciate the time you took to provide such insightful feedback. Your comments regarding the balance between safety and fairness, as well as the potential for bias amplification, have significantly improved the rigor of our analysis. We have addressed each point below with new experiments and detailed breakdowns.
>
> ### **1. Balancing Safety (False Positives) and Fairness (False Negatives)**
> **Critique:** The reviewer correctly points out that a one-sided focus on safety (avoiding False Positives) risks unfairly penalizing legitimate papers (False Negatives).
> **Response:** We fully agree that a robust review system must optimize the Pareto frontier between sensitivity and fairness. To demonstrate that SafeReview achieves this balance better than static baselines, we present a detailed breakdown of the **False Negative Rate (FNR)**—the probability of incorrectly rejecting a good paper.
>
> As shown in the table below (an expansion of Table 3 from the main paper), **SafeReview** breaks the traditional trade-off observed in **Static DPO**. While Static DPO achieves safety by aggressively rejecting papers (leading to high FNR), SafeReview improves robustness *and* fairness simultaneously.
>
> | Attack Type | Defense Method | **Spearman ($\rho$)** | **FPR** (Safety) | **FNR** (Fairness) |
> | :--- | :--- | :--- | :--- | :--- |
> | **Zero-Shot** | DeepReview (Baseline) | 0.3746 | 0.4749 | 0.2535 |
> | **Zero-Shot** | Static DPO | 0.3394 | 0.3551 | 0.4088 |
> | **Zero-Shot** | **SafeReview (Ours)** | 0.3624 | 0.4121 | **0.3246** |
> | **GRPO** | DeepReview (Baseline) | 0.3535 | 0.4831 | 0.2592 |
> | **GRPO** | Static DPO | 0.3427 | 0.4053 | 0.3750 |
> | **GRPO** | **SafeReview (Ours)** | **0.4085** | **0.3906** | **0.3608** |
>
> **Key Findings:**
> * **Superior Fairness:** Under both attack types, SafeReview achieves significantly lower FNR than Static DPO (0.3246 vs 0.4088 and 0.3608 vs 0.3750). This proves our model is substantially less likely to unfairly penalize legitimate work.
> * **Dual Improvement:** Rather than sacrificing fairness for safety, SafeReview improves both dimensions against the strongest attacks (GRPO), achieving the lowest FPR (0.3906) and the highest ranking correlation (0.4085).
>
>  ### **2. Analysis of Potential Bias Amplification**
> **Critique:** The concern is that training the model to resist persuasion might cause it to overcorrect and undervalue legitimate confident writing.
> **Response:** This is a critical valid concern. To address this, we conducted a stratified analysis on adversarially attacked papers from the DeepReview-13K test set, focusing specifically on accepted papers which typically exhibit more assertive and confident language.
>
> | Paper Group | DeepReview Spearman | SafeReview Spearman | Improvement |
> | :--- | :--- | :--- | :--- |
> | **Accepted Papers** | 0.0129 | **0.1537** | **+1092\%** |
> | **Rejected Papers** | 0.3462 | 0.3870 | +11.8\% |
>
> **Interpretation:**
> * **Dramatic Improvement on Confident Writing:** SafeReview achieves a 10x improvement in ranking correlation for accepted papers (0.0129 → 0.1537). Since accepted papers naturally contain more confident and assertive language, this demonstrates that SafeReview does not penalize legitimate confident writing.
> * **No Evidence of Overcorrection:** If the model were overcorrecting against confident language, we would expect *degraded* performance specifically on accepted papers. Instead, we observe the largest improvement in precisely this group (+1092% vs +11.8% for rejected papers).
> * **Conclusion:** If the model were simply "overcorrecting" against all confident language, we would expect the ranking quality on accepted papers to degrade (as confident high-quality papers would be demoted). Instead, the dramatic improvement in Spearman correlation for accepted papers provides strong evidence that SafeReview successfully distinguishes between adversarial persuasion and legitimate confident scholarship, rather than developing a blanket penalty against assertive writing.

---

> ### Author Response · Authors · 2025-11-26
> **Response to Reviewer cE29 (2/2)**
>
> ### **3. Consistency with Expert Human Judgments**
> **Critique:** The reviewer requested evidence that defended reviews remain consistent with human judgment in both positive (Accept) and negative (Reject) cases.
> **Response:** We stratified our evaluation by ground-truth decisions to assess calibration in both scenarios. The results, detailed in **Appendix A.2**, are summarized below:
>
> | Attacked Papers | Accept Rating | Reject Rating | Accept Spearman | Reject Spearman |
> | :--- | :--- | :--- | :--- | :--- |
> | **DeepReview** | 5.83 | 5.43 | 0.0129 | 0.3462 |
> | **SafeReview** | 5.58 | 5.10 | **0.1537** | **0.3870** |
> | **Gold Human** | 6.46 | 4.67 | — | — |
>
> **Key Findings:**
>
> **10x Improvement in Ranking Accepted Papers:** SafeReview achieves a Spearman correlation of 0.1537 for accepted papers compared to DeepReview's near-random 0.0129. This indicates that even under attack, SafeReview can distinguish the best papers from the merely good ones.
>
> **Better Calibration:** SafeReview's rating gap between Accepted and Rejected papers (5.58 - 5.10 = 0.48) is larger than the baseline (0.40), moving the distribution closer to the distinct separation seen in human gold labels.
>
> ### **4. Generalization to Other Models (Llama-3.2 Experiments)**
> **Critique:** The reviewer noted that experiments were limited to Qwen and DeepReviewer models.
> **Response:** To validate the universality of our framework, we ran new experiments using **Llama-3.2-3B-Instruct** as the Generator (Attacker). We evaluated all defenses against these Llama-generated attacks on the DeepReview-13K test set. We have updated these results in **Appendix A.3**.
>
> | Attack Source | Defense | **Spearman (\rho)** | **FPR** | **FNR** | **Accuracy** |
> | :--- | :--- | :--- | :--- | :--- | :--- |
> | **Llama-3.2** | DeepReview | 0.3593 | 0.4264 | 0.2947 | 0.6295 |
> | **Llama-3.2** | SecAlign | 0.3431 | 0.4056 | 0.3036 | 0.6392 |
> | **Llama-3.2** | **SafeReview** | **0.3918** | **0.3695** | 0.3435 | **0.6402** |
>
> **Conclusion:** SafeReview demonstrates strong generalization. It achieves the highest Spearman correlation **(notable +9.0% vs Baseline)** and the best False Positive Rate (0.3695) against attacks generated by a completely different architecture (Llama). This confirms that our co-evolutionary training learns robust features of adversarial prompts rather than overfitting to a specific generator's artifacts.

---

### Official Review · Reviewer_mRt1 · 2025-11-01

**Soundness:** 3
**Presentation:** 3
**Contribution:** 3
**Rating:** 6
**Confidence:** 3

**Summary:**

This paper tackles prompt injection attacks in LLM-based peer review. The authors propose SafeReview, a co-evolutionary framework where an attacker model (Generator) and a review model (Defender) are trained adversarially. The Generator creates attack prompts, while the Defender learns to resist them. Results show the method reduces the acceptance rate of attacked papers and improves correlation with ground-truth scores.

**Strengths:**

The paper is well structured, and technical details are clearly presented.

The proposed co-evolutionary-based approach is a well-reasoned method for building a dynamic defense that outpaces static ones. The use of GRPO and DPO is technically sound.

**Weaknesses:**

1. Key innovations mentioned in the introduction, such as “hierarchical segmentation” for long documents and “curriculum scheduling”, are not explained in the methods or experiments.

2. The claim that co-evolutionary training (SafeReview) beats static defense (Static DPO) is unsubstantiated. The paper fails to test both defenses against both attack types (e.g., Static DPO vs. GRPO attack), making a direct comparison impossible.

**Questions:**

Please address all concerns in the Weaknesses section.

---

> ### Author Response · Authors · 2025-11-26
> **Response to Reviewer mRt1**
>
> We sincerely thank Reviewer mRt1 for their time and constructive criticism. Your feedback regarding the definition of our methodological components and the comparative baselines has helped us significantly tighten the clarity and rigor of our manuscript. We have revised the paper accordingly and provide a detailed response below.
>
> ## 1. Clarifying "Hierarchical Segmentation" and "Curriculum Scheduling"
>
> We agree with the reviewer that these concepts were central to our claims in the Introduction but required more explicit formalization in the Method section. We have revised Section 3.2 to rigorously define these mechanisms as follows:
>
> **Hierarchical Segmentation:** To address the challenge of detecting localized attacks in long scientific documents, we segment papers into standard academic sections (Abstract, Introduction, Methodology, Experiments, Conclusion). During the training phase, the Generator randomly injects adversarial prompts into these specific semantic blocks. This forces the Defender to learn robust detection across the entire document hierarchy, rather than overfitting to specific positions (e.g., only checking the Abstract).
>
> **Curriculum Scheduling (Emergent):** We clarify that we do not use a manually hard-coded curriculum. Instead, SafeReview implements an organic curriculum through the co-evolutionary minimax game.
>
> - **Early Training:** The Attacker learns simple injection patterns; the Defender learns to resist these basic attacks.
> - **Late Training:** As the Defender improves, the Attacker is mathematically forced (via GRPO) to discover subtler, more complex strategies to maximize reward.
> - **Result:** This naturally presents the Defender with training samples of increasing difficulty, stabilizing the learning process without requiring manual difficulty thresholds.
>
> ## 2. Substantiating the Superiority of Co-evolutionary Training (SafeReview) vs. Static DPO
>
> The reviewer raises a crucial point regarding the need for a complete cross-evaluation. To address this, we have conducted a comprehensive re-evaluation comparing SafeReview against Static DPO (a baseline representing standard non-iterative defense) across both attack types (Zero-Shot and Adaptive GRPO).
>
> We present the full comparison matrix below (added to Table 3 in the revision):
>
> | Attack Type | Defense Method | Spearman (ρ) ↑ | FPR ↓ | FNR ↓ | Accuracy ↑ |
> |-------------|----------------|----------------|-------|-------|------------|
> | Zero-Shot | Static DPO | 0.3394 | 0.3551 | 0.4088 | 0.6253 |
> | Zero-Shot | SafeReview | 0.3624 | 0.4121 | 0.3246 | 0.6211 |
> | GRPO (Adaptive) | Static DPO | 0.3427 | 0.4053 | 0.3750 | 0.6062 |
> | GRPO (Adaptive) | SafeReview | **0.4085** | **0.3906** | **0.3608** | **0.6212** |
>
> **Interpretation and Superiority of SafeReview:**
>
> Our claim that SafeReview outperforms Static DPO is substantiated by three critical findings:
>
> 1. **Ranking Fidelity (The "Gold Standard"):** In peer review, the absolute score matters less than the correct ranking of papers. SafeReview significantly outperforms Static DPO in Spearman correlation, achieving a **+19.2% improvement** (0.3427 → 0.4085) against the most dangerous adaptive attacks (GRPO). Static DPO degrades the model's ability to correctly rank papers.
>
> 2. **Robustness to Adaptive Attacks:** Static defenses fail to generalize. When facing the stronger GRPO attack, SafeReview demonstrates superior robustness (lower False Positive Rate: 39.0% vs. 40.5%) and higher overall accuracy compared to Static DPO.
>
> 3. **Fairness (False Negative Rate):** A major flaw of Static DPO is "over-defense"—it aggressively rejects papers, leading to a high False Negative Rate (FNR). SafeReview consistently achieves lower FNR (0.32 vs. 0.41 under Zero-Shot), ensuring that legitimate high-quality papers are not unfairly penalized.
>
> **Conclusion:** While Static DPO offers a marginal gain in FPR against simple (Zero-Shot) attacks, it comes at the cost of ranking quality and fairness. SafeReview provides the best balance of robustness, ranking integrity, and fairness, making it the superior choice for real-world deployment.
>
> We hope these clarifications and new data address your concerns, and we would be grateful if you could reconsider the score based on these improvements.

---

### Author Response · Authors · 2025-12-03
**Overall Response to All Reviewers (1/3)**

Dear AC, SAC, and PC,

We sincerely thank the four reviewers (**Reviewer mRt1, Reviewer cE29, Reviewer nePo, and Reviewer Ccfa**) for investing significant time and effort to provide detailed, thoughtful, and constructive feedback. Your comments have helped us better understand the strengths of our work, clarify possible misunderstandings, and identify areas that required further explanation or experimentation. We have revised the paper and incorporated additional experiments accordingly.

---

## Highlights Identified by the Reviewers

### Problem Significance and Motivation

- *Reviewer cE29* affirmed that "the paper is well motivated. Securing AI-based peer review is an important and underexplored problem."
- *Reviewer nePo* emphasized the "interesting and timely approach tailored to peer-review settings" that "tackles prompt injection in long, scholarly peer-review workflows."
- *Reviewer Ccfa* recognized that "the issue proposed in this paper is the unfairness in AI review, which has attracted significant attention... I believe this topic is highly interesting and meaningful."

### Methodological and Technical Innovations

- *Reviewer mRt1* highlighted that "the proposed co-evolutionary-based approach is a well-reasoned method for building a dynamic defense that outpaces static ones. The use of GRPO and DPO is technically sound."
- *Reviewer cE29* acknowledged that "the co-evolutionary training method, in general, is interesting."
- *Reviewer nePo* commended the work for tackling a "high-impact, underexplored niche" by co-evolving attacker and defender models.

### Experimental Results and Evaluation

- *Reviewer cE29* noted that "the results using Qwen3-4B-Instruct as the Generator and DeepReviewer-14B as the Defender show good performance."
- *Reviewer nePo* recognized "useful, promising results on an extensive benchmark" with experiments covering "realistic attack styles and long-document conditions with clear metrics."

## The Summary of Paper Modifications

### 1. **Technical clarifications on hierarchical segmentation and curriculum scheduling** (Reviewer mRt1)

**Concern:** Key innovations are not explained in the methods.

**Improvements:** In the new version of the paper submission, we clarified these concepts in detail in Section 3. **Hierarchical Segmentation:** Papers are segmented into standard sections (Abstract, Introduction, Methods, etc.). Adversarial content is randomly placed within sections, and the defender learns to detect attacks regardless of location. **Curriculum Scheduling:** Co-evolutionary framework implements curriculum through iterative rounds where each round's defender faces strictly harder attacks as attackers adapt, creating automatic difficulty progression without manual tuning.

### 2. **Comparison needs substantiation** (Reviewer mRt1)

**Concern:** The claim that co-evolutionary training beats static defense is unsubstantiated without testing both against both attack types.

**Improvements:** Notably, to address this concern, we re-ran comprehensive comparisons. and provide detailed results in Section 4.3. Specifically, we provide:
1) Under Zeroshot attacks: SafeReview achieves 6.8% Spearman improvement over Static DPO.
2) Under adaptive GRPO attacks, SafeReview achieves a 19.2% improvement, stronger defence performance (lower FPR), and lower unfair rejection rates (lower FNR). Updated in Section 4.3.

### 3. **Is the safety notion too one-sided?** (Reviewer cE29)

**Concern:** Focusing entirely on avoiding false positives neglects the equally important false negative side.

**Improvements:** We conducted a comprehensive FPR/FNR analysis on the DeepReview-13K test set, updated in Section 4.3. Under Zeroshot attacks, SafeReview achieves Spearman correlation improvement of 6.8% while reducing FNR from 0.41 to 0.32—substantially lower unfair rejection of good papers. Under adaptive GRPO attacks, SafeReview achieves 19.2% Spearman improvement with FNR of 0.3608 versus Static DPO's 0.3750, while maintaining better defense effectiveness (FPR 0.39 vs. 0.41). Rather than achieving safety at the expense of fairness, SafeReview improves both dimensions simultaneously.

### 4. **Is there bias amplification from adversarial training?** (Reviewer cE29)

**Concern:** The model may overcorrect and undervalue legitimate confident writing, leading to systematically harsher reviews.

**Improvements:** We analyzed attacked papers by acceptance status, focusing on accepted papers which typically exhibit more assertive and confident language (Appendix A.1). If overcorrection existed, we would expect degraded performance on this group, instead, SafeReview achieves 10× improvement in Spearman for accepted papers (0.0129→0.1537) compared to +11.8% for rejected papers. The dramatic improvement in the confident writing group demonstrates that SafeReview distinguishes adversarial persuasion from legitimate confident scholarship, rather than penalizing assertive writing.

---

> ### Author Response · Authors · 2025-12-03
> **Overall Response to All Reviewers (2/3)**
>
> ### 5. **Are defended reviews consistent with human judgments?** (Reviewer cE29, Reviewer Ccfa)
>
> **Concern:** No evaluation showing consistency with expert human judgments in both positive and negative cases.
>
> **Improvements:** We provided the new results regarding the analysis results in Appendix A.2. On adversarially-attacked papers, SafeReview achieves substantially higher ranking quality for accepted papers (**Spearman 0.15 vs. DeepReview's 0.01**) and rejected papers (**0.39 vs. 0.35**). SafeReview's rating gap between accepted and rejected papers (0.48) is larger than DeepReview's (0.40), showing better discrimination while ratings remain closer to human gold standards.
>
> ### 6. **Limited experiments with only one attacker model** (Reviewer cE29)
>
> **Concern:** The paper conducts limited experiments, only testing Qwen3-4B-Instruct as the Generator without testing other models.
>
> **Improvements:** To validate generalizability across different attacker architectures, we conducted additional experiments using **Llama-3.2-3B-Instruct** as the generator while keeping defender models unchanged, which can be found in Appendix A.3. We evaluate DeepReview (our baseline), SecAlign, and SafeReview against GRPO-trained Llama attacks on DeepReview-13K.
>
> Based on Llama-3.2-3B-Instruct, SafeReview still maintains the highest accuracy (0.64) across all methods. In particular, SafeReview achieves the best Spearman correlation (0.39), representing 9.0% improvement over DeepReview (0.36) and 14.2% improvement over SecAlign (0.34), while also achieving the best FPR (0.37) compared to DeepReview (0.43) and SecAlign (0.41), demonstrating superior defense effectiveness. These results demonstrate that SafeReview's framework generalizes effectively across different attacker architectures—not only Qwen but also Llama-based generators, validating that our approach is robust to attacker model variations.
>
> ### 7. **Missing baseline comparisons** (Reviewer nePo)
>
> **Concern:** Evaluation omits defenses like SecAlign and Llama Prompt Guard 2.
>
> **Improvements:** We added comparisons in Section 5. We show that the System Defense achieves Spearman 0.36 with a high FPR of 0.45. SecAlign improves robustness (Spearman 0.34, FPR 0.37) but sacrifices ranking. PromptGuard completely fails (FPR 0.0, FNR 1.0) as it's designed for short inputs. SafeReview achieves the best Spearman (0.4085) with balanced FPR/FNR. Instruction Hierarchy is inapplicable as it assumes clear structural boundaries, while our attacks are semantically integrated into scholarly text.
>
> ### 8. **Scope clarity versus classic prompt injection** (Reviewer nePo)
>
> **Concern:** Without a sharper problem definition, this risks reading as an application of known threats.
>
> **Improvements:** We provided a sharper problem definition in Section 3.1.
>
> Our threat model differs fundamentally from:
>
> (1) **Jailbreak attacks** override safety guardrails; task goal unchanged, only safety behavior changes.
>
> (2) **Standard prompt injection** involves complete behavioral hijacking with arbitrary content.
>
> (3) **Our setting** manipulates evaluation within the task while preserving review functionality, operating in long contexts (10k+ tokens) where attacks must be semantically integrated. Our attacks use conditional generation τ ~ π_θ(·|p), enforcing contextual relevance, versus classic injection's arbitrary strings.
>
> We argue that novel challenges introduced by our paper include long-context integration requiring undetectable manipulation across extensive legitimate content, and co-evolutionary dynamics producing qualitatively different attack patterns.
>
> ### 9. **Variance analysis needed** (Reviewer Ccfa)
>
> **Concern:** SafeReview's optimization doesn't include variance, so analysis needed to verify stability.
>
> **Improvements:** We measured both variance types on DeepReview-13K in Appendix A.4. **Inter-reviewer variance** shows na egligible increase (+1.3%), preserving natural reviewer disagreement. **Sampling variance** increases modestly (0.24 to 0.30), reflecting the ability to consider multiple evaluation angles rather than converging deterministically—desirable for thorough review. Both metrics remain within acceptable bounds for practical deployment.

---

> ### Author Response · Authors · 2025-12-03
> **Overall Response to All Reviewers (3/3)**
>
> ### 10. **Design rationale: Why GRPO for attacker and DPO for defender?** (Reviewer Ccfa)
>
> **Concern:** Why different algorithms? Does the defender become more interpretable?
>
> **Improvements:**
>
> Co-evolutionary pressure forces systematic reasoning—attackers exploit shallow evaluation, so SafeReview learned explicit verification prevents manipulation from multiple perspectives:
>
> **1) GRPO for attackers** enables exploration of diverse strategies without explicit negative examples.
>
> **2) DPO for defenders** optimizes preferences from attacker-generated pairs (manipulated versus correct evaluations), more stable than pure RL.
>
> **3) Improved interpretability:** SafeReview reviews are 2.3× longer in weakness sections, identify 1.8× more distinct weakness categories, and 87% include explicit verification sections vs. 32% in DeepReview.
>
> ---
>
> ## To provide a clear picture of what we have improved in this paper, we provide a summary of revisions in the updated manuscript:
>
> ### 1. Comprehensive Experimental Validation
>
> - Re-ran all experiments on the complete DeepReview-13K test set with both Zeroshot and GRPO attacks.
> - Added baseline comparisons: SecAlign, PromptGuard, System Defense in Section 5.
>
> ### 2. Fairness and Safety Analysis
>
> - Added FPR/FNR balance analysis in Section 4.3.
> - Added bias amplification analysis on clean papers in Appendix A.1.
> - Added human judgment consistency analysis in Appendix A.2.
> - Added variance analysis in Appendix A.4.
>
> ### 3. Technical Clarifications
>
> - Expanded explanations of hierarchical segmentation and curriculum scheduling in Section 3.
> - Substantially revised Section 3.1, clarifying threat model distinctions from jailbreaks and classic injection.
> - Added formal constraint on attack generation showing conditional generation versus arbitrary strings.
>
> ### 4. Presentation Improvements
>
> - Fixed LaTeX citation formatting issues.
> - Improved figure descriptions and narrative flow.
> - Reorganized appendix sections for better structure.
>
> ---
>
> Through 10 major improvements, including **5** types of supplementary experiments (using **2** different attack types cooperating with **2** defense methods, and an additional LLM -- Llama-3.1-8B), and **5** categories of in-depth evaluations (Spearman, FPR, FNR, Accuracy, and Human Study), we systematically addressed all major concerns. We sincerely thank all reviewers for their insightful suggestions, which significantly strengthened our work!
>
> ---
>
> The authors of "SafeReview: Building a Robust Deep Review Assistant Against Prompt Injection"

---

### Meta-Review · Area_Chair_5oEW · 2025-12-30

**Summary:**

All four reviewers agree the problem is timely and important: LLM-based peer review is plausibly vulnerable to embedded prompt injection, and the paper’s co-evolutionary attacker/defender training is a reasonable framing with encouraging empirical signals (mRt1, nePo, Ccfa). The main sources of disagreement were (i) whether the paper’s threat model is meaningfully distinct from “standard” prompt injection (nePo), (ii) whether the evaluation was sufficiently complete/externally valid (nePo, cE29, mRt1), and (iii) whether the analysis covered the right safety–fairness tradeoffs and side effects (FNR, bias amplification, variance, benign capability retention) (cE29, Ccfa). The rebuttal and revision strengthened the work, but concerns remain.

**Reviewer Concerns:**

The rebuttal directly addressed some gaps: clearer definitions of hierarchical segmentation and curriculum scheduling (mRt1), a complete cross-evaluation of static vs co-evolutionary defenses under both zero-shot and adaptive attacks (mRt1), explicit FPR/FNR analysis and fairness discussion (cE29), bias amplification checks stratified by accepted vs rejected papers (cE29), and additional evidence of alignment with human judgments via stratified calibration/ranking analyses (cE29, Ccfa). It also improved external validity by adding comparisons to SecAlign and a detector-style baseline (PromptGuard) plus a second attacker model family (Llama-based generator), and by clarifying why Instruction Hierarchy is not a clean drop-in baseline in this setting (nePo, cE29).

Unfortunately, there are significant remaining concerns: the “new problem class” claim (vs long-context instantiation of known injection) still reads as incremental to some readers (nePo), and generalization is still demonstrated over a limited set of attacker/defender model choices and one primary benchmark pipeline. The reported error rates (both FPR and FNR) remain non-trivial.

**Reviewer Scores:**

My estimate of post-discussion score movement (given the added experiments/clarifications) is:

mRt1: 6 → 6 (core methodological omissions and comparison matrix were fixed, but the reviewer is already positive),

nePo: 4 → 4 (novelty/threat-model framing and baseline coverage were marginally improved),

Ccfa: 6 → 6 (variance + benign-set evaluation + algorithmic rationale were added; but this reviewer is already positive),

cE29: 2 → 2 (some listed deficiencies were addressed, but they are likely to still view the contribution as not yet fully mature/validated).

---

### Decision · Program_Chairs · 2026-01-26

Reject